# A Pilot Feasibility Study on the Use of Dual-Joystick-Operated Ride-on Toys in Upper Extremity Rehabilitation for Children with Unilateral Cerebral Palsy

**DOI:** 10.3390/children11040408

**Published:** 2024-03-29

**Authors:** Sudha Srinivasan, Patrick D. Kumavor, Kristin Morgan

**Affiliations:** 1Physical Therapy Program, Department of Kinesiology, University of Connecticut, Storrs, CT 06269, USA; 2Institute for Collaboration on Health, Intervention, and Policy (InCHIP), University of Connecticut, Storrs, CT 06269, USA; 3The Institute for the Brain and Cognitive Sciences (IBACS), University of Connecticut, Storrs, CT 06269, USA; 4Biomedical Engineering Department, University of Connecticut, Storrs, CT 06269, USA; patrick.d.kumavor@uconn.edu (P.D.K.); kristin.2.morgan@uconn.edu (K.M.)

**Keywords:** joystick-operated ride-on toys, bimanual training, engagement, unilateral cerebral palsy, upper extremity, constraint-induced movement therapy, technology-based aids, rehabilitation, therapy adjunct, feasibility study

## Abstract

Children with unilateral cerebral palsy (UCP) require task-oriented practice several hours per week to produce meaningful gains in affected upper extremity (UE) motor function. Clinicians find it challenging to provide services at the required intensity and sustain child engagement. This pilot study assessed the acceptance and utility of a child-friendly program using dual-joystick-operated ride-on toys incorporated into an intensive UE rehabilitation camp. Eleven children with UCP between four and 10 years received ride-on-toy navigation training for 20–30 min/day, five days/week, for three weeks as part of camp programming. We report session adherence and percent time children spent in task-appropriate attention/engagement across sessions. The overall effects of camp programming on children’s motor function were assessed using the Shriner’s Hospital Upper Extremity Evaluation (SHUEE) from pretest to posttest and using training-specific measures of bimanual UE use and navigational accuracy. Children showed excellent adherence and sustained task-appropriate engagement across sessions. The combined program led to improved navigational accuracy (*p*-values ≤ 0.007) as well as spontaneous affected UE use during bimanual activities outside the training context (*p* < 0.001). Our pilot study provides promising evidence for using modified, commercially available ride-on toys to incentivize rehabilitation and boost repetitive, task-oriented UE practice among children with UCP.

## 1. Introduction

Among contemporary approaches for upper extremity (UE) training in children with unilateral cerebral palsy (UCP) or hemiplegia, constraint-induced movement therapy (CIMT) and bimanual training (BT) have emerged as showing the strongest evidence for improving functional UE outcomes [1,2]. Constraint-based training approaches focus on encouraging forced use of the affected UE by physically restraining the unaffected UE [3]. Although highly successful in promoting spontaneous use and motor function on the affected side, this approach does not afford opportunities for training bimanual coordination [4]. Bimanual coordination is foundational for achieving independence in daily activities such as buttoning clothes, tying laces, opening bottles, using scissors, carrying a loaded lunch tray, etc. [4,5]. Therefore, hybrid models that combine CIMT and BT have been implemented to promote spontaneous use and train motor skills in the affected UE in isolation and in conjunction with the unaffected UE during goal-directed bimanual functional tasks [5,6,7,8,9]. For example, a 90 h, two-week intensive hybrid summer camp (6 h/day CIMT and 2 h/day BT) led to improvements in fine motor skills, specifically, pinch strength, in 23 children with UCP between 12 and 18 years [8]. Similarly, another two-week hybrid UE training protocol led to improved functional skills and independence in daily activities in young children with UCP compared to conventional rehabilitation [10]. Overall, there is considerable evidence for task-oriented training approaches combining forced affected UE use and bimanual training in children with hemiplegia. 

Irrespective of the type of training paradigm, improvements in affected UE function are related to high-intensity treatment dosing (60–90 h) [1,3,11,12,13]. Typically, this high dosage of task-oriented training is provided in intensive bursts within a group-based camp model lasting for a few weeks [12,14]. Given the intensive and challenging nature of the training, it is critical that training activities be designed to maximize child engagement and buy-in [15,16]. Motivation is a powerful mediator of neuroplastic brain changes [16,17,18]. Therapeutic contexts that are fun and novel, incorporate choices, provide social support, and invoke a sense of mastery and competence in the child promote motivation and sustained engagement [19,20,21,22]. Sustained engagement during therapy, in turn, can encourage children to persevere through challenging activities and foster repetitive practice needed for motor learning, and is associated with favorable therapeutic outcomes [16,19,23]. 

To make intensive training more engaging, researchers have explored several solutions, including themed camps (e.g., circus-based or magic training) and the use of technology-based playful solutions such as virtual reality, robotic exoskeletons, and exergames as part of existing training programs [16,24,25]. Such innovative therapy ideas can complement conventional therapy effectively and may offer several advantages: They include intrinsically motivating activities that break the monotony of traditional training protocols, extend therapy opportunities beyond clinic-based settings, foster repetitive UE practice, provide immediate feedback on movement performance, and help improve children’s adherence to therapy. For example, a recent study suggests that a 30 h, eight-day, group-based, personalized upper limb therapy program that combined CIMT, BT, and exergame-based robotics led to improvements in the performance of daily activities and the dexterity of the affected and unaffected UEs, as well as the achievement of individualized goals in 23 children with hemiplegia between 4 and 12 years [6]. Similarly, five hours of training using an assistive exoskeleton and virtual reality as part of a two-week, 60 h CIMT camp led to clinically significant improvements in bimanual skills and occupational performance that were sustained at 6-month follow-up [26]. Overall, several innovative therapy adjuncts have been evaluated for their complementary role in fostering therapy engagement and boosting training dosing in children with hemiplegia. 

Our group has been exploring the use of joystick-operated ride-on toys as a technology-based, age appropriate, and child-friendly therapy adjunct that can be used to incentivize use of the affected UE in children with hemiplegia [27,28,29]. This paper reports findings from a pilot study exploring the acceptability and utility of a training program using dual-joystick-operated ride-on toys incorporated as part of a three-week hybrid training camp for children with hemiplegia. Our choice of a bimanual training program using dual-joystick-operated toys stems from evidence that bimanual function is strongly predictive of improvements in occupational performance among children with hemiplegia [30,31]. Our study had two aims: (1) to assess participant acceptability of incorporating dual joystick-operated ride-on-toy navigation training (DJT) as part of an intensive hybrid training camp in children with hemiplegia and (2) to report the combined effects of the overall training program (DJT + other camp activities) on children’s bimanual motor function measured on standardized and training-specific measures. We hypothesize that the DJT program will be acceptable for children, as indicated by high levels of adherence and attentional engagement during training-related activities. Moreover, the overall training program will lead to improvements in children’s affected UE use and bimanual function both during ride-on-toy training sessions as well as on a standardized test of motor performance outside the training context.

## 2. Materials and Methods

### 2.1. Participants and Setting 

The single-group, quasi-experimental, pretest–posttest study was conducted within the premises of a local high school. Participants were recruited through convenience sampling from a camp for children with hemiplegia called the Lefty and Righty Camp of Connecticut. The camp director shared information about the study with camp participants, and interested families contacted the research team. Study inclusion criteria were children between four and ten years with hemiplegia/clear asymmetry in UE function. We included children across all levels of impairment, as assessed using the Manual Ability Classification System (MACS). The MACS classifies children’s use of their hands to handle objects during daily activities on a five-point scale, with lower scores indicating better abilities [32]. Study exclusion criteria included children with recent UE trauma or surgery within the previous six months, uncorrected visual impairments, inability to maintain supported sitting for at least 20 min, or children who exceeded the weight limit of the ride-on toys (>150 lb). The study was approved by the Institutional Review Board of the University of Connecticut (protocol #: H21-0019). We obtained parental permission from all families as well as verbal/written assent from all children. Table 1 reports demographic information of the study participants. 

### 2.2. Procedures 

#### 2.2.1. Camp Structure and Program

Children participated in the camp for six hours/day, five days/week, over three weeks. The camp staff included physical therapists, occupational therapists, and paraprofessional staff. Each child was paired with an aide who worked one-on-one with the child under the supervision of a licensed clinician. The camp followed a hybrid model that incorporated principles of CIMT and BT [3,7,8]. The camp was structured to incorporate both unimanual and bimanual UE task practice. Children practiced unimanual tasks with their affected arm for roughly five hours/day, with their unaffected arm constrained in a cast (based on principles of CIMT). Children also received one hour/day of BT, during which their cast was removed. Both types of practice (unimanual and bimanual) focused on promoting gross and fine motor skills, muscle strength, hand dexterity, and UE range of motion. The ride-on-toy training sessions were provided as part of the children’s daily bimanual practice. The camp incorporated child-friendly themes each day, including animated movies and sport-specific themes to maintain child engagement and motivation across training weeks. 

#### 2.2.2. Dual-Joystick-Operated Ride-on-Toy Navigation Training (DJT) Program

We used modified, commercially available, joystick-operated ride-on toys, i.e., the Wild Thing^TM^ (FisherPrice) and Huffy Green Machine Vortex (Huffy) for our training protocol (see Figure 1; note that the Wild Thing was used with younger children and the Huffy Green Machine was used with older children). The ride-on toys required the children to use both hands in a coordinated manner to drive the toy (i.e., push both joysticks simultaneously to move forward, pull back with both hands together to move backward, push with right joystick and/or pull back with left joystick to turn left and vice versa to turn right). The commercial toys were modified by the research team to (a) improve access to the joysticks by raising their height, adding forearm support plates and adding foam balls on top of the joystick to aid grasp; (b) provide external support through a built-up support structure created through PVC pipes; and (c) improve comfort and postural support by adding foot plates (to the Huffy) and providing additional trunk support (using pool noodles, kickboards) as needed by the child (see Figure 1). All researchers delivering the intervention sessions were physical therapists with experience working with children with cerebral palsy. As appropriate, we facilitated optimal alignment of the child’s trunk through verbal cues to encourage the children to actively correct their position, tactile cues on the child’s trunk to facilitate weight shifts and more symmetrical weight bearing, and, if needed, manual, therapist-initiated repositioning of the child’s body to facilitate symmetry, optimal alignment, and postural control.

The ride-on-toy navigation training program was provided every day at camp for 20–25 min/session. The training had the following key ingredients: (a) multidirectional navigation and (b) structured practice of functional UE skills involving reach, grasp, manipulation, and release. Multidirectional navigational games involved obstacle courses of incremental complexity (e.g., straight, arc, and slalom paths) that required the children to skillfully maneuver the toy using both hands per task demands, including driving forward and backward; turning 90, 180, and 360 degrees to the right and left; starting and stopping; changing directions; and driving on different surfaces (flat and inclined surfaces). Structured UE practice involved repetitive practice of functional UE movement patterns that children typically are required to engage in during activities of daily living, such as reaching in different directions, throwing–catching, push–pull activities, precision and power grasps/grips (e.g., opening–closing jars, envelopes, picking up large and small objects), and release of objects. The UE practice was focused on training the affected UE for the role of a mobilizer as well as a stabilizer during bimanual tasks. We used principles of motor learning, including encouraging variable practice, discovery learning, and active problem solving, providing immediate feedback and reinforcement on performance, employing a least-to-most prompting hierarchy, and tailoring the program to provide the “just right” challenge for each child [33]. 

### 2.3. Measures 

We report on two types of measures from this study: acceptability metrics and measures of UE motor function. The acceptability metrics were designed to assess the practicality/feasibility of incorporating DJT into an intensive hybrid training program to improve UE function in children with UCP. We also used a combination of standardized tests and training-specific measures to assess the combined efficacy of DJT and hybrid task-oriented, intensive UE training in improving motor function in children with hemiplegia. All testing and training sessions were videotaped to allow for further coding. For each of the standardized and training-specific measures, two independent coders (undergraduate students) trained by the study principal investigator, i.e., first author coded a subset of the videos (20%) twice to establish inter- and intra-rater reliability of over 90%. All discrepancies between coders were resolved through discussions with the first author and consensus building. 

#### 2.3.1. Acceptability Metrics

Adherence: We tracked children’s participation in the DJT sessions over the course of the three-week program. Adherence was monitored through daily researcher-maintained logs that documented completion of the session, session duration, child affect, and any other observations/comments from researchers. We report on individual and group data from children on adherence rates for the training program. 

Attentional engagement: We coded videos of early and late DJT sessions using a behavioral coding software called Datavyu (v.1.3.8; https://datavyu.org/) to evaluate the children’s attentional focus during the training. The children’s attention during the session was categorized as being (a) task-directed (child is looking at the path they are traversing or at the props/objects that they are interacting with during the UE tasks), (b) socially directed (child is looking at trainers, clinicians, peers, etc.), (c) directed towards device handling (child is looking at the joystick–hand interface or parts of the toy that are related to moving the toy, such as the wheels), and (d) task-inappropriate (child is distracted, zoned out, looking elsewhere, or not engaged in the training activities). We were interested in the proportion of task-directed attention that we used as a proxy for the children’s level of engagement with the training activities. 

#### 2.3.2. Measures of Motor Function

Standardized test: We used the Shriner’s Hospital Upper Extremity Evaluation (SHUEE) [34] as a test of bimanual motor function. The SHUEE is a video-based test comprising 16 bimanual tasks. The test assesses the quality of children’s spontaneous use of their affected UE while performing 9 of these 16 tasks (called spontaneous functional analysis, or SFA). The SFA is scored on a six-point scale ranging from no use of the affected UE in either a stabilizing or a mobilizing role (score 0) to spontaneous and optimal use of the affected UE for task performance (score of 5). Additionally, all 16 tasks are scored on the dynamic segmental alignment of the affected UE at the elbow, forearm, wrist (sagittal and frontal planes), thumb, and fingers (called dynamic positional analysis, or DPA). This scoring is completed on a four-point scale ranging from pathological alignment (score of 0) to typical alignment (score of 3). During performance of SHUEE tasks, we also scored the extent of independent/unprompted use of the affected UE by the children (0: child needed multiple prompts from the tester to use their affected arm beyond initial instructions, 1: child needed a single prompt from the tester to use their affected arm beyond the initial instructions, 2: child used affected arm without prompting from the tester). We also tracked the types of prompts required by the children during each task: gestural (e.g., tester points at the affected arm to encourage its use), verbal (e.g., tester uses verbal cues such as “use righty/lefty” to encourage affected UE use), or manual (e.g., tester provides hand-on-hand assistance to get the child to use their affected UE). We calculated a weighted prompt score per test item, with manual prompts receiving the highest score (weight of three) compared to verbal (weight of two) and gestural (weight of one) prompts. A higher prompt score indicated that the child required a variety of prompts to complete the task. The maximum possible prompt score per task was six. We summed the prompt score across all tasks of the SHUEE to calculate a total raw prompt score (maximum possible prompt score of 96). We report on percent total SFA and DPA scores [35,36,37,38], percent unprompted affected UE use, and raw prompt scores. At present, there are no data available on standard error of measurement or minimal clinically important difference for the SHUEE.

Training-specific measures: We used videos from early and late DJT sessions to code metrics representative of navigation accuracy and bimanual arm use during ride-on-toy training sessions. Early and late DJT sessions involved the children traveling along a set of pre-defined, incrementally challenging paths (i.e., straight, arc, and slalom paths) that were demarcated using painter’s tape and cups/cones. We coded the number of times that the children went outside these demarcated paths during the session. Moreover, we also tracked the number of times the children bumped against fixed (e.g., wall, door, cabinet, etc.) or movable (e.g., cups, cones, bowling pins, etc.) objects/obstacles during the training sessions. We report on the frequencies in standard time of out-of-path episodes as well as bumps against fixed and moving objects. To assess bimanual arm use, we coded the navigational bouts within the session (i.e., parts of the session when the child was driving the toy) and coded the percent time during navigation spent in bimanual activity (i.e., bouts when both arms were active simultaneously to control the joysticks), unimanual activity (i.e., only the unaffected or the affected arm was pushing/pulling the joystick), or no-activity bouts (i.e., child stopped during navigation due to fatigue or poor muscular control). 

### 2.4. Statistical Analyses

All data were assessed for assumptions of parametric statistics. We assessed normality using the Kolmogrov–Smirnov test of normality. Among our dependent variables, data on SHUEE, obstacle contacts, and navigational accuracy satisfied assumptions of normality. Some of the dependent variables related to children’s attentional patterns and bimanual arm use did not satisfy the assumptions of normality. For all variables that violated the assumptions of normality, we ran analyses using non-parametric Wilcoxon signed rank tests. The results obtained from non-parametric tests were similar to the major trends observed using parametric statistics for attentional patterns and bimanual UE use. Given the ease of interpretation of measures of central tendency and variability associated with parametric statistics, we report on data obtained from parametric statistics in the Results section. We used repeated measures analyses of variance (ANOVAs) and dependent *t* tests to analyze intervention-related changes in acceptability- and motor function-related measures. If the Mauchly test of sphericity was violated for the ANOVAs, then Greenhouse Geisser corrections were applied. For the ANOVA analyzing attentional states, we used attention type (task-directed, socially directed, device-handling, and task-inappropriate) and time (early and late sessions) as our within-subjects factors. Similarly, to analyze data on bimanual UE use, we used UE activity during navigation (bimanual, affected UE only, unaffected UE only, no UE activity) and time (early and late sessions) as the within-subjects factors in the ANOVA. Post-hoc testing for significant main and interaction effects of ANOVAs were conducted using dependent *t* tests. In cases where there was a significant main effect and an interaction involving the same factors, we analyzed the interaction effect only. We used dependent *t* tests to analyze intervention effects on outcome measures associated with the SHUEE (%SFA, %DPA, %unprompted affected UE use, and raw prompt scores) and navigational accuracy (frequencies in standard time of out-of-path deviations, bumps against fixed objects, and bumps against movable objects). We set significance at *p* ≤ 0.05. We applied Bonferroni corrections to account for type II error for all analyses. Given the pilot nature of our study, we report on both group data as well individual trends. Moreover, we also report on effect sizes (ES) for significant intervention-related effects using standardized mean difference values [39,40]. We calculated ES as posttest–pretest or late–early session differences and report on absolute values of the ES, where higher values indicate improvements in assessed outcomes. The standardized mean difference values were classified as small (0.2–0.49), medium (0.5–0.79), or large (≥0.8) [41]. 

## 3. Results

### 3.1. Acceptability Metrics

#### 3.1.1. Adherence

Average adherence was 96.10%. Two children missed one session each and two children missed two sessions each due to sickness, scheduling conflicts, or holiday plans. Children drove the toy for an average of 20 min (mean (SE)—20.80(0.87); range: 17.86 to 27.21 min; see Table 2 for details). None of the children refused to participate in the DJT sessions. Table 2 provides individual data on acceptability metrics. 

#### 3.1.2. Attentional Engagement

The repeated measures ANOVA indicated a significant main effect of attention type (*F* (1.486, 14.855) = 217.983, *p* < 0.001, *ηp*^2^ = 0.956). Post hoc analyses suggested that irrespective of session, the children directed significantly greater duration of attention to task-appropriate targets (mean (SE): 73.87 (2.55) compared to all other types of attentional targets, with the least amount of time spent being distracted (socially directed: mean (SE): 15.36 (2.31); device-handling: mean (SE): 8.03 (1.26); task-inappropriate: mean (SE): 2.74 (0.49)). These data suggest sustained engagement with training activities across early and late sessions of the program (see Figure 2 for a session-wise breakdown of attentional engagement and Table 2 for individual data from participants).

### 3.2. Measures of Motor Function

#### 3.2.1. Standardized Test 

Dependent *t* tests conducted on outcome variables associated with the SHUEE suggested a significant improvement from pretest to posttest in percent total SFA scores (*t*(10) = −6.547, *p* < 0.001) and percent unprompted affected UE use (*t*(10) = −4.359, *p* = 0.001) but not in raw prompt scores (*t*(10) = 1.998, *p* = 0.074) or percent total DPA scores (*t*(10) = −0.994, *p* = 0.344). Specifically, 11 out of 11 children showed medium-sized improvements in %SFA scores (ES = 0.63), and 9 out of 11 children demonstrated medium-sized increases (ES = 0.69) in percent independent, unprompted use of their UE during all tasks (see Figure 3 and Table 3). 

#### 3.2.2. Training-Specific Measures 

Dependent *t* tests on variables associated with navigational accuracy/control suggested a significant decrease from early to late sessions in bumps against fixed objects (*t*(10) = 4.566, *p* < 0.001) and out-of-path deviations (*t*(10) = 3.343, *p* = 0.007) but not bumps against moveable objects (*t*(10) = 0.391, *p* = 0.704). Specifically, 11 out of 11 children showed a large effect size (ES = 0.91) for decrease in bumps against fixed objects, and 9 out of 11 children showed a medium-sized decrease (ES = 0.70) in out-of-path deviations from early to late training sessions (see Figure 4 and Table 4 for details).

In terms of bimanual UE use, the repeated measures ANOVA indicated a significant main effect of arm activity (*F* (1.113, 11.132) = 237.290, *p* < 0.001, *ηp*^2^ = 0.960) and a significant arm activity × time interaction effect (*F* (1.177, 11.769) = 8.335, *p* = 0.011, *ηp*^2^ = 0.455). Post hoc analyses of the activity × time interaction suggested that overall, across early and late sessions, the children spent a significantly greater percentage of time engaged in bimanual versus unimanual and no-activity bouts (see Figure 5 and Table 4 for details). Moreover, from early to late sessions, children showed a medium-sized increase (ES = 0.78) in percent time in the session spent in bimanual activity, with a concurrent medium-sized decrease (ES = 0.57 to 0.75) in time spent in unimanual activity (see Figure 5, *p*-values < 0.05). All 11 children followed the group trends (see Table 4). 

## 4. Discussion

Our pilot study suggests that it was feasible to incorporate DJT as part of a hybrid group-based UE training camp for children with UCP. We found high levels of adherence and sustained task-direction attention towards training activities. Moreover, the combined hybrid camp program inclusive of DJT led to medium-to-large improvements in affected UE motor function during bimanual tasks as measured on a standardized test from pretest to posttest and during ride-on-toy navigation training from early to late sessions. Overall, dual-joystick-operated ride-on toys seem to be well-accepted therapy adjuncts that can be feasibly incorporated into current intensive training models for children with hemiplegia to boost therapy engagement and improve spontaneous affected UE use during bimanual tasks.

Evidence-based guidelines suggest that 30–60 h of goal-directed practice is required to improve UE motor ability in children with hemiplegia [12,42]. However, in most clinical settings, children receive weekly therapy for 30 min to 2 h, making it challenging to provide the required dosing of task-oriented practice [43,44]. There is a need for “drastic rearrangement of therapy models” to ensure that children receive the optimal dosing of high-intensity UE practice [42]. Some potential solutions include intensive training camps or home programs to supplement clinic-based care [42]. A related issue that impacts therapy dosing is child motivation [20,45]. Per the self-determination theory, individuals are likely to persist with activities that are (a) aligned with their interests and provide a sense of choice/control (autonomy), (b) allow for connection with social partners (relatedness), and (c) afford a sense of mastery and confidence (competence) [19,20,45,46]. Consistent with this theoretical framework, recent clinical practice guidelines in CP recommend that chosen therapeutic activities be enjoyable and intrinsically motivating for children, tailored to individual needs and preferences, and involve practice within children’s naturalistic environment to maximize carryover to real-world contexts [14].

Our training program was based on principles of motivational and motor learning theories. We chose ride-on devices because they are popular, intrinsically rewarding, and inclusive toys that promote exploratory play and self-initiated bimanual UE use [47,48,49]. Our training program fostered child autonomy, provided children the “just-right” bimanual motoric challenge and immediate multimodal feedback (visual, auditory, tactile) on their actions, and challenged the children’s perceptual, action-related, and motor planning/cognitive skills [27,28,50]. The use of ride-on toys as a therapy adjunct is also aligned with qualitative perspectives shared by children and caregivers that technology and gamification help break the monotony of traditional therapy, fosters greater compliance, promotes sustained engagement during challenging UE activities, and provides a feasible alternative to conventional exercises while fostering repetitive practice [51]. 

We found that DJT in combination with other camp activities led to improvements in unprompted spontaneous affected UE use during bimanual tasks both within and outside the training context. The training might have led to increased awareness of the affected UE, prompting greater spontaneous use [24]. High levels of child engagement across sessions may have motivated the children to attempt and persist with bimanual tasks that may have been avoided previously. The program may have impacted the children’s mastery motivation, i.e., their drive to independently persist with difficult tasks with a goal to achieve mastery [30,31]. Increased practice of using the affected UE in a variety of roles (mobilizer and stabilizer) during camp activities may have contributed to improved functional arm strength, leading to increased spontaneous use of the affected UE. As the children experienced success and improved their UE skills, e.g., while driving the toy, they may have persisted longer despite the increase in challenge levels of the training tasks. 

Our findings are in line with other studies that used technology-based tools such as virtual reality, augmented reality, and virtual games/exergames to incentivize UE goal-directed practice [6,52,53,54,55,56,57,58,59,60,61,62]. For example, Hung and colleagues assessed the feasibility and preliminary efficacy of a 12-week (24-session) training protocol using a suite of Kinect-based unilateral and bilateral UE motion-controlled games in 13 children with CP. Similar to our study, they found high adherence and enjoyment during Kinect-based training, and, in combination with conventional occupational therapy, children improved UE motor function on standardized tests from pretest to posttest [60]. Similarly, a six-week randomized controlled trial with 36 children with hemiplegia suggested that compared to the control group that only received conventional exercise, the experimental group that received conventional exercise + virtual reality gamified exercise training demonstrated significantly greater improvements in affected arm range of motion and function on standardized tests, spontaneous use during daily activities as indicated by a parent questionnaire, and improved overall quality of life following the intervention, with gains maintained at three-month follow-up [55]. Overall, this study adds to the growing literature in support of the use of innovative and intrinsically rewarding tools/technologies in conjunction with conventional rehabilitation to increase child engagement, boost therapy dosing, and promote motor learning among children with hemiplegia. 

### 4.1. Limitations 

Our study is limited by a small convenience sample, lack of a control group, heterogeneity in symptoms of the included participants, and lack of follow-up tests following completion of the camp. Given our single-group design, we were unable to systematically test the efficacy of the DJT program for children with hemiplegia. Moreover, the reported improvements in motor outcomes may be attributed to camp programming that comprised activities based on CIMT, BT, and DJT. We are unable to measure the isolated effects of DJT at this time. However, we will use these pilot data to estimate effect sizes and design an adequately powered, two-group, larger randomized clinical trial that will examine the differential effects of DJT compared to dose-matched conventional bimanual therapy provided to children with hemiplegia. 

### 4.2. Clinical Implications

Children with hemiplegia frequently fail to achieve the optimal dosing required for neuroplasticity-driven changes in motor function due to the scarcity of motivating interventions that can incentivize goal-directed affected UE use both within and outside therapy sessions [15]. Motivation and engagement are “active ingredients” that promote repetitive practice and sustained effort during rehabilitation interventions [63]. Participation in engaging activities can stimulate neural areas underlying attention, motor planning, and cognition, ultimately leading to memory consolidation [15,63]. There is evidence that children with CP seem to be more motivated by technology-/game-based interventions compared to conventional rehabilitation [15,18]. Our research therefore focused on assessing the feasibility and efficacy of the therapeutic use of modified, commercially available, joystick-operated ride-on toys with children with hemiplegia. 

In the past, we demonstrated the feasibility and positive preliminary effects of single-joystick-operated ride-on toys (controls provided on the child’s affected side) implemented within a modified CIMT camp [27,28]. Given the importance of bimanual skills in daily life, in this study, we incorporated ride-on-toy navigation training using dual joystick controls within a hybrid UE training program to promote bimanual coordination. The DJT program was feasible to implement within the hybrid camp protocol and was well received by the children, caregivers, and clinicians [64]. The preliminary findings from this study in conjunction with our past work suggest that training using ride-on toys seems to create an engaging environment for children and may increase child willingness to persist in intensive practice involving the affected UE. Our future studies will involve controlled clinical trials to assess the short-term and long-term adjunctive effects of ride-on-toy navigation training delivered by clinicians and caregivers in a variety of real-world settings with children with hemiplegia. 

## 5. Conclusions

Our pilot study evaluated the acceptance and utility of incorporating a technology-based training program using modified commercially available joystick-operated ride-on toys within an intensive upper extremity rehabilitation camp in 11 children with UCP. As part of the camp activities, the children received a structured, incrementally challenging program using dual-joystick-operated ride-on toys for 20–30 min/day, five days/week for three weeks. Children showed high levels of adherence and sustained task-appropriate attention/engagement across training sessions. The program was seamlessly integrated into existing camp programming. The comprehensive camp-based program inclusive of ride-on-toy navigation training led to improvements in spontaneous use of the affected upper extremity during navigation as well as outside the training context during a bimanual test of motor performance. Our preliminary findings add to recent literature on the use of technology-based therapeutic adjuncts to facilitate client buy-in and promote goal-directed practice of purposeful and intrinsically rewarding activities with the affected UE. Our future work will systematically assess the efficacy of clinic- and home-based training programs using ride-on toys compared to dose-matched standard-of-care training programs to promote UE motor function among children with hemiplegia. 

## Figures and Tables

**Figure 1 children-11-00408-f001:**
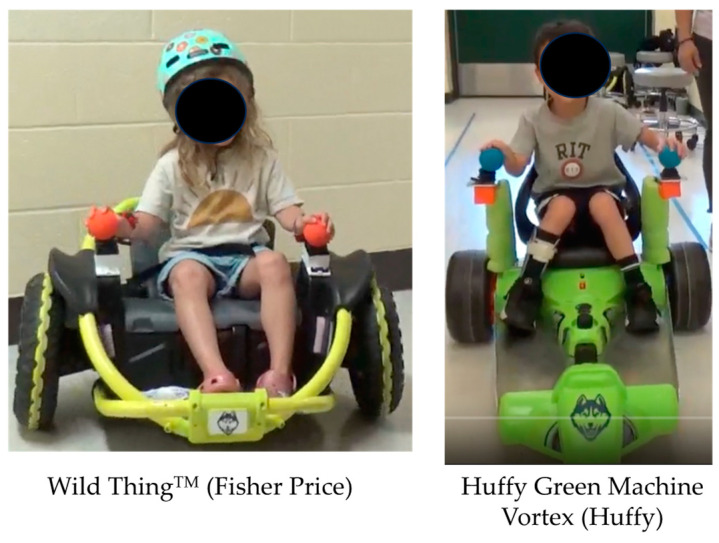
Ride-on toys used in the study.

**Figure 2 children-11-00408-f002:**
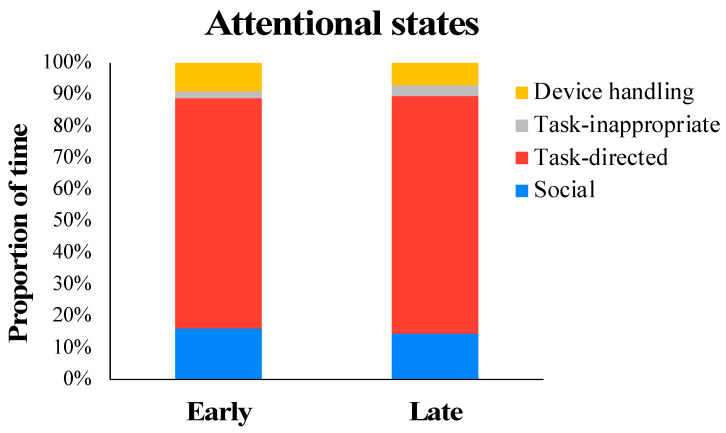
Attentional engagement of the children during DJT sessions.

**Figure 3 children-11-00408-f003:**
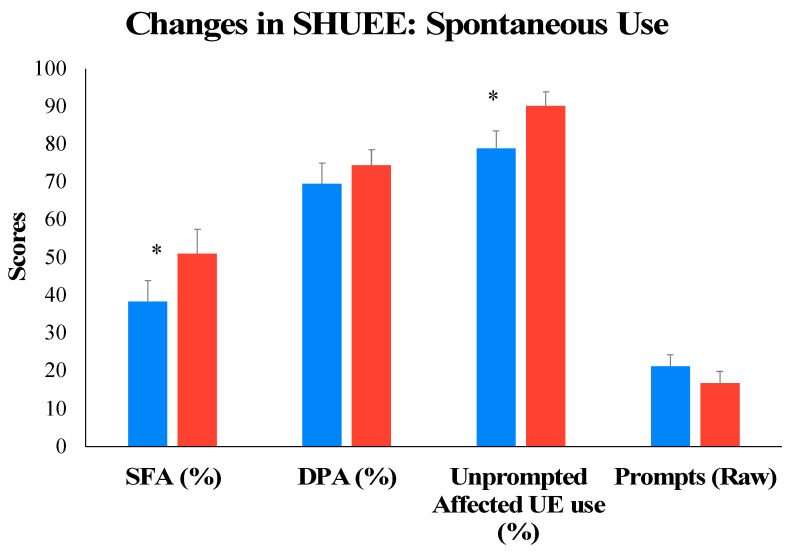
Training-related changes in motor function on the standardized SHUEE from pretest to posttest. * *p* ≤0.05.

**Figure 4 children-11-00408-f004:**
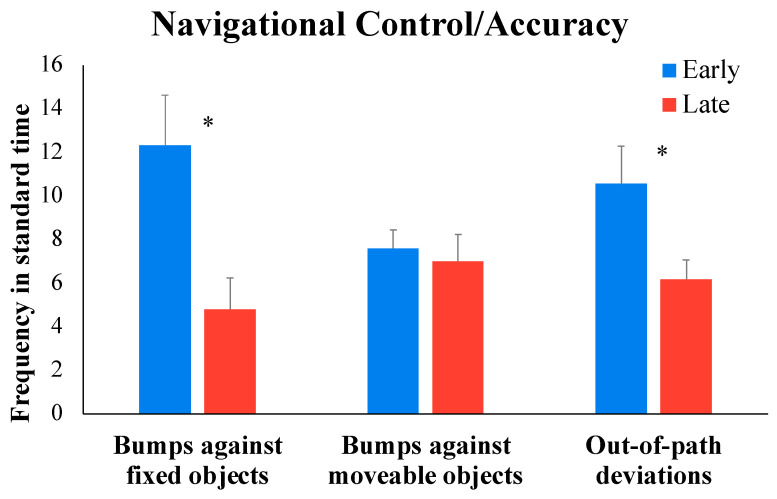
Training-related changes in navigational accuracy from early to late DJT sessions. * *p*
≤0.05.

**Figure 5 children-11-00408-f005:**
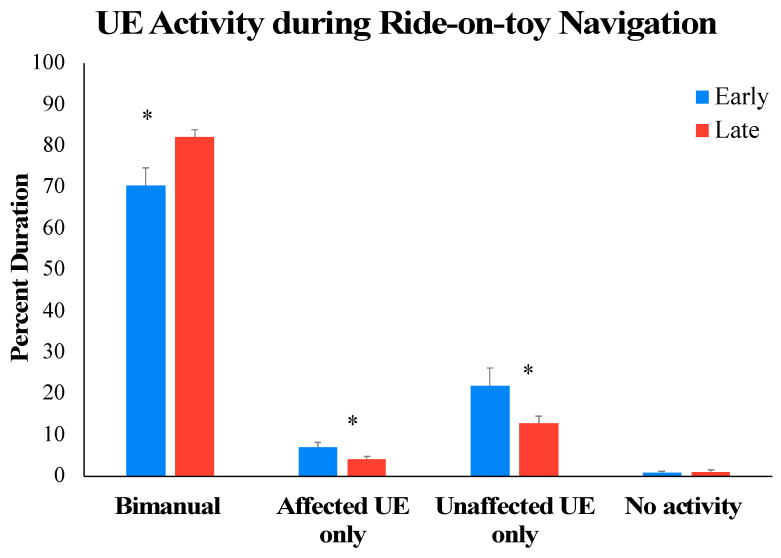
Training-related changes in bimanual UE use from early to late DJT sessions. * *p*
≤0.05.

**Table 1 children-11-00408-t001:** Demographic details of the participants in the study.

Demographic Information	Age in Years	Gender	Race/Ethnicity	Side of Weakness	MACS Score
Child 1	4.45	F	White/Non-Hispanic	L	II
Child 2	9.93	M	White/Non-Hispanic	R	II
Child 3	5.50	F	White/Non-Hispanic	L	II
Child 4	6.10	M	Multiracial—Korean, Irish, Polish, Puerto Rican	R	II
Child 5	10.19	F	White/Hispanic	R	II
Child 6	7.91	F	White/Non-Hispanic	R	II
Child 7	5.19	M	White/Non-Hispanic	L	III
Child 8	7.92	M	Mixed—White, Asian	R	I
Child 9	5.05	F	White/Non-Hispanic	L	II
Child 10	4.35	F	White/Non-Hispanic	L	I
Child 11	4.93	F	White/Non-Hispanic	R	III

F: female; M: male; R: right, L: left; MACS: Manual Ability Classification System. MACS Score: Level I—Handles objects easily and successfully, Level II—Handles most objects but with somewhat reduced quality and/or speed of achievement, Level III—Handles objects with difficulty; needs help to prepare and/or modify activities [32].

**Table 2 children-11-00408-t002:** Individual data on training adherence and attentional patterns during early and late training sessions.

Acceptability Metrics	Adherence (%)	Average Session Time (min)	Task-Related Attention (%)		Social Attention (%)		Device Handling (%)		Elsewhere (%)	
			Early	Late	Early	Late	Early	Late	Early	Late
Child 1	100.00	20.71	80.66	60.19	12.24	28.72	6.40	7.37	0.69	3.72
Child 2	100.00	19.36	85.14	82.35	9.06	14.19	5.16	2.95	0.64	0.51
Child 3	92.86	17.15	72.19	77.58	14.76	10.24	11.51	5.07	1.54	7.11
Child 4	100.00	21.36	71.15	64.94	19.30	29.30	8.51	4.34	1.04	1.42
Child 5	85.71	20.17	73.90	82.70	6.93	13.25	18.55	3.57	0.63	0.47
Child 6	85.71	23.75	84.60	79.55	8.86	14.13	4.21	2.73	2.32	3.59
Child 7	92.86	22.08	61.87	51.75	20.16	16.12	16.87	26.11	1.11	6.03
Child 8	100.00	20.93	84.55	81.65	7.44	9.83	7.31	2.28	0.71	6.24
Child 9	100.00	27.21	69.93	83.37	10.99	7.26	11.12	7.44	7.97	1.93
Child 10	100.00	17.86	38.73	80.36	55.16	7.18	3.74	10.18	2.37	2.28
Child 11	100.00	18.21	78.17	79.81	12.36	10.43	5.42	5.84	4.05	3.91

Note: All attention data are represented in percent duration of total session dedicated to attention to different targets.

**Table 3 children-11-00408-t003:** Individual data on outcome variables associated with the standardized SHUEE.

SHUEE Variables	% SFA		% DPA		% Unprompted Affected UE Use		Raw Prompt Score	
	Pretest	Posttest	Pretest	Posttest	Pretest	Posttest	Pretest	Posttest
Child 1	51.11	62.22	77.78	75.00	87.50	96.88	19.00	12.00
Child 2	28.89	37.78	69.44	69.44	78.13	90.63	28.00	25.00
Child 3	35.56	57.78	86.11	86.11	75.00	96.88	24.00	19.00
Child 4	42.22	51.11	75.00	76.39	62.50	90.63	34.00	17.00
Child 5	20.00	24.44	58.33	44.44	81.25	81.25	10.00	14.00
Child 6	37.78	57.78	52.78	63.89	71.88	87.50	27.00	20.00
Child 7	6.67	8.89	40.28	61.11	46.88	59.38	38.00	40.00
Child 8	66.67	77.78	86.11	83.33	100.00	100.00	6.00	6.00
Child 9	53.33	71.11	93.06	80.56	87.50	100.00	21.00	4.00
Child 10	60.00	75.56	84.72	93.06	96.88	100.00	10.00	10.00
Child 11	20.00	37.78	41.67	86.11	81.25	90.63	17.00	19.00

SHUEE: Shriner’s Hospital Upper Extremity Evaluation; SFA: spontaneous functional analysis; DPA: dynamic positional analysis; UE: upper extremity.

**Table 4 children-11-00408-t004:** Individual data on training-specific variables collected during ride-on-toy navigation training sessions.

Training-Specific Variables	Measures of Navigational Control/Accuracy						Bimanual UE Use During Navigation							
	Bumps—Fixed Obstacles		Bumps—Moveable Obstacles		Out-of-Path Deviations		Bimanual Use		Affected UE Only		Unaffected UE Only		No UE Activity	
	Early	Late	Early	Late	Early	Late	Early	Late	Early	Late	Early	Late	Early	Late
Child 1	10.93	2.22	7.24	6.61	16.36	11.08	73.77	88.80	8.65	5.12	17.25	6.08	0.33	0.00
Child 2	3.74	1.51	3.76	16.43	5.26	2.86	78.32	86.46	7.63	1.40	12.13	9.81	1.93	2.33
Child 3	14.34	3.15	5.32	7.32	9.96	4.02	79.91	89.53	7.59	2.95	11.80	7.52	0.70	0.00
Child 4	14.70	5.65	6.62	6.24	10.99	3.30	65.93	75.46	7.67	1.69	24.54	17.90	1.87	4.96
Child 5	4.03	0.00	7.99	2.01	1.66	3.76	85.57	86.08	2.46	5.39	11.78	8.53	0.19	0.00
Child 6	24.41	5.81	7.15	8.10	11.99	6.01	80.72	84.50	4.32	2.43	14.79	11.43	0.17	1.64
Child 7	5.00	4.00	13.78	10.97	9.09	7.78	36.73	78.47	0.86	0.96	59.80	20.57	2.61	0.00
Child 8	9.52	0.00	4.76	1.35	1.60	4.01	77.89	78.24	5.71	9.56	16.24	11.37	0.16	0.83
Child 9	25.85	13.18	9.96	5.76	15.53	10.70	56.66	82.75	6.85	3.71	35.66	13.02	0.84	0.52
Child 10	15.78	14.19	10.55	6.51	15.69	8.73	74.58	82.47	10.86	5.53	14.05	11.45	0.51	0.55
Child 11	7.27	3.05	6.31	5.61	18.03	5.61	63.35	71.82	14.29	4.98	21.98	23.20	0.38	0.00

UE: upper extremity. Note: Bumps and out-of-path deviations are reported in frequency per standard time. Bimanual UE use metrics are reported as percent duration of navigation time spent in bimanual/unimanual/no-activity bouts.

## Data Availability

The data presented in this study are available on request from the corresponding author. The data are not publicly available due to specific ethical and privacy considerations.

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
