# Peer review of "A Pilot Feasibility Study on the Use of Dual-Joystick-Operated Ride-on Toys in Upper Extremity Rehabilitation for Children with Unilateral Cerebral Palsy"

_children, 2024, doi:10.3390/children11040408_

Round 1
Reviewer 1 Report
Comments and Suggestions for Authors
Peer review of:
Title: A pilot feasibility study on the use of dual joystick-operated ride-on-toys in upper extremity rehabilitation for children with Unilateral Cerebral Palsy
Journal: Children
Dear authors
1. In my opinion, this was a worthwhile study because the high intensity and lengthy duration of rehabilitation programmes that are necessary to help children with cerebral palsy to improve upper limb use and function is well-proven, and remains an obstacle. I think the authors document this evidence satisfactorily, but some references are inappropriate for this purpose. For example, reference 63 is a case study and these prove nothing at all. I hope that the following points will help the authors with the manuscript and in future research.
2. It is customary in manuscripts to spell numbers from one to nine, and then use Arabic digits from 10 onwards.
3. The study had two aims: (1) to assess participant acceptability of incorporating Dual Joystick-operated ride-on-toy navigation Training (DJT) as part of an intensive hybrid training camp in children with hemiplegia and (2) to assess the combined effects of the overall training program (DJT + other camp activities) on children’s bimanual motor. The authors hypothesised that the DJT program would be acceptable for children as indicated by high levels of adherence and attentional engagement with training-related activities and that the overall training program would improve children’s affected upper limb use and bimanual function both during ride-on-toy training sessions as well as demonstrated by a standardized test of motor performance.
4. Essentially this was feasibility study to assess whether Dual Joystick-operated ride-on-toys, which looked a lot of fun, were acceptable to the children and showed high levels of adherence and attentional engagement. But note that you can’t test hypotheses without a control group, which this study did not include. I suggest that the training programme included many hours of CIMT and BT, which is now supported by plenty of evidence, and so improvements could be expected if the children engaged in the programme. I note that you have included references to your previous work, which I investigated, and found similar errors in that work e.g. claiming that the intervention had shown improvements with no control group, and inappropriate use of statistical tests (see paragraph 9 below) and summary statistics as in paragraph 6 below. It is wrong to suggest that something is ‘trending’ towards the result that you wish to see if the statistical tests are non-significant.
5. I was delighted that the children did show improvements, but it is wrong throughout the manuscript to try and suggest that this was down to the combined programme, given that there is no control group which comprised a randomly-allocated group of children with cerebral palsy that shared the same characteristics as the intervention group but did not participate in the combined programme.
6. Note that it is wrong to calculate the mean and SE of ordinal data. Ordinal categories may each have a number allocated, but these are simply denoting their position in their ordered state: a label, not a real number. As an example, if you had 12 children with one at MACS Level 5, three at MACS level 4, six at MACS level 3 and two at MACS level 2, the ‘mean’ MACS level 2 would be MACS level 3.25. This level doesn’t actually exist and the result of the calculation is actually meaningless. But in any case, mathematical operations on ordinal data are not allowed.
7. With only 11 children, why did you decide against tabulating their characteristics and also provide all outcome data in tables as well. It is not possible to check the results if the data aren’t provided. For example, the range of MACS levels could be observed, min and max values for adherence examined, and do on. Please tabulate the results so that they can be investigated, and include any negative results.
8. I am unable to determine how you came to 72 as the total score for total raw prompt scores, this needs explaining please. Note that this will be ordinal data as well.
9. Note that the raw scores for the SHUEE are ordinal, and so the ANOVA and t-tests are not appropriate. What is the MCID for the SHUEE? A statistically-significant result is one thing, and although the ES is suggestive, these have been calculated as if the raw scores were interval-level data which they are not, and the MCID is important to determine if there was a difference that mattered to the children and their parents.
10. Can you explain why you used percent total SFA scores? I don’t understand this or what it means.
I hope these comments will be taken as constructively as intended. I think there is mileage in intensive training camps such as the one for which you have demonstrated feasibility of incorporating Dual Joystick-operated ride-on-toy navigation Training. The benefits of such an inclusion remain to be demonstrated, and require an appropriate methodology that includes two or more randomly-allocated groups that are adequately-powered for the primary outcome measure.
Author Response
We thank the reviewer for their thoughtful comments.
Children manuscript titled: “A pilot feasibility study on the use of dual joystick-operated ride-on-toys in upper extremity rehabilitation for children with Unilateral Cerebral Palsy”
Response to reviewers
We thank the reviewers for their insightful comments and critiques. We have revised the manuscript significantly in accordance with reviewer suggestions. We think the paper is significantly strengthened following these edits. We hope that the reviewers find our edits satisfactory. Please find below a point-by-point response to reviewer comments. All manuscript edits are using track changes. Thank you very much for your time and we look forward to hearing from the journal.
Reviewer 1:
Comment 1: In my opinion, this was a worthwhile study because the high intensity and lengthy duration of rehabilitation programmes that are necessary to help children with cerebral palsy to improve upper limb use and function is well-proven, and remains an obstacle. I think the authors document this evidence satisfactorily, but some references are inappropriate for this purpose. For example, reference 63 is a case study and these prove nothing at all. I hope that the following points will help the authors with the manuscript and in future research.
Response: We thank the reviewer for their feedback regarding the need and merit of our study. We understand the reviewer’s point about the strength of the references. We have removed the case study from our list of references and have instead cited recent review papers and studies that employed larger samples. We thank the reviewer for all of their feedback on our manuscript and our line of research.
Comment 2: It is customary in manuscripts to spell numbers from one to nine, and then use Arabic digits from 10 onwards.
Response: We thank the reviewer for their suggestion. We have edited the manuscript to reflect the suggested changes.
Comment 3: The study had two aims: (1) to assess participant acceptability of incorporating Dual Joystick-operated ride-on-toy navigation Training (DJT) as part of an intensive hybrid training camp in children with hemiplegia and (2) to assess the combined effects of the overall training program (DJT + other camp activities) on children’s bimanual motor skills. The authors hypothesised that the DJT program would be acceptable for children as indicated by high levels of adherence and attentional engagement with training-related activities and that the overall training program would improve children’s affected upper limb use and bimanual function both during ride-on-toy training sessions as well as demonstrated by a standardized test of motor performance. Essentially this was feasibility study to assess whether Dual Joystick-operated ride-on-toys, which looked a lot of fun, were acceptable to the children and showed high levels of adherence and attentional engagement. But note that you can’t test hypotheses without a control group, which this study did not include. I suggest that the training programme included many hours of CIMT and BT, which is now supported by plenty of evidence, and so improvements could be expected if the children engaged in the programme. I note that you have included references to your previous work, which I investigated, and found similar errors in that work e.g. claiming that the intervention had shown improvements with no control group, and inappropriate use of statistical tests (see paragraph 9 below) and summary statistics as in paragraph 6 below. It is wrong to suggest that something is ‘trending’ towards the result that you wish to see if the statistical tests are non-significant.
Response: We understand the limitations of our study design. Our project was an unfunded, pilot feasibility study to assess the acceptability and utility of a training program using dual joystick-operated ride-on-toys to serve as an adjunct to conventional therapy that is aimed at improving upper extremity function in children with CP. The preliminary effects reported in the study are a result of the camp programming that includes CIMT, BT, and DJT. We agree with the reviewer that it is not possible at this time for us to understand the isolated effects of DJT in children with CP. We have acknowledged this limitation within the limitations section of the manuscript. Although not reported in this paper, stakeholders (i.e., caregivers, clinicians, and children) reported on exit questionnaires that DJT was very enjoyable for children (reported by 91-100% of all stakeholder groups). Moreover, caregivers and clinicians provided feedback that engaging in DJT as part of other camp activities led to increase in children’s awareness of their affected arm and motivation to use their affected upper extremity both during and outside the navigation training sessions (please note that 8 out of the 11 children in the camp had participated in the summer camp over multiple previous years; although our exit questionnaires did not ask this explicitly, caregivers in their open-ended responses frequently compared their experience of camp programming in that year inclusive of DJT with previous years when the DJT experience was not included within the camp. Their responses suggested that caregivers recognized the added value of incorporating DJT into the camp and all caregivers and clinicians asked for the ride-on-toy training to be incorporated again into camp during the following year). Clinicians and caregivers shared their perspectives that use of an age-appropriate and fun activity was associated with children’s greater self-initiated use of their affected upper extremity, increased confidence, and willingness to repeat training activities (these data are reported in a publication that is currently in advanced stages of review in a different journal). Taken altogether, the promising acceptability and feasibility metrics from this study have helped inform our next set of efficacy trials that will compare the DJT program with a dose-matched program based on conventional therapy. The justification for use of statistics is provided in responses to comments 8 and 9.
Comment 4: I was delighted that the children did show improvements, but it is wrong throughout the manuscript to try and suggest that this was down to the combined programme, given that there is no control group which comprised a randomly-allocated group of children with cerebral palsy that shared the same characteristics as the intervention group but did not participate in the combined programme.
Response: We understand the reviewer’s point. As mentioned in the response to comment 3, this was an unfunded pilot feasibility study that was designed to assess acceptance, utility, and feasibility of implementation/integration of the DJT program into a conventional therapy program. We have cautioned readers regarding the interpretation of our study findings in light of design limitations in the revised manuscript. Please see limitations section of the revised manuscript (lines 1191-1198, page 13).
Comment 5: Note that it is wrong to calculate the mean and SE of ordinal data. Ordinal categories may each have a number allocated, but these are simply denoting their position in their ordered state: a label, not a real number. As an example, if you had 12 children with one at MACS Level 5, three at MACS level 4, six at MACS level 3 and two at MACS level 2, the ‘mean’ MACS level 2 would be MACS level 3.25. This level doesn’t actually exist and the result of the calculation is actually meaningless. But in any case, mathematical operations on ordinal data are not allowed.
Response: We understand the reviewer’s point. In the revised manuscript, we have created table 1 to report on MACS levels across each of the 11 participants in the study.
Comment 6: With only 11 children, why did you decide against tabulating their characteristics and also provide all outcome data in tables as well. It is not possible to check the results if the data aren’t provided. For example, the range of MACS levels could be observed, min and max values for adherence examined, and do on. Please tabulate the results so that they can be investigated, and include any negative results.
Response: We thank the reviewer for their excellent suggestion. In the revised manuscript, we have added Table 1 that provides individual data on participant demographics. Moreover, we have also added tables that provide data from individual children for the outcomes reported within the manuscript (Tables 2-4) to allow the reader to evaluate individual level data. We hope that these edits will address the reviewer’s comment.
Comment 7: I am unable to determine how you came to 72 as the total score for total raw prompt scores, this needs explaining please. Note that this will be ordinal data as well.
Response: We apologize for the typo in the manuscript. The maximum possible prompt score is 96. We have corrected this error in the revised version of the manuscript. We calculated a weighted prompt score per test item of the SHUEE with manual prompts receiving the highest score (weight of 3) compared to verbal (weight of 2) and gestural (weight of 1) prompts. A higher prompt score indicated that the child required a variety of prompts to complete the task. The maximum possible prompt score per task was 6. We summed the prompt score across all tasks of the SHUEE (16 tasks with maximum possible prompt score per task being 6) to calculate a total raw prompt score, with maximum possible total prompt score across all items of the SHUEE being 96. We have explained our rationale for using parametric statistics to analyze outcome measures related to the SHUEE in response to comment 9. However, please note that our results were similar upon conducting non-parametric Wilcoxon signed rank tests on our outcome measures for the SHUEE.
Comment 9: Can you explain why you used percent total SFA scores? I don’t understand this or what it means.
Response: We thank the reviewer for their question. The SFA scores of the SHUEE have a range from 0 (does not use affected upper extremity) to 5 (spontaneous use, partial to complete use of affected upper extremity). The SFA is scored for 9 out of the 16 tasks. Therefore, the range of SHUEE scores extends from 0 (minimum score) to 45 (maximum score). To ease reader’s interpretation of SHUEE score, we converted the raw total SFA scores to a percent SFA score. For example, a child with a SHUEE SFA score of 18 will receive a percent total SFA score of 40. Higher scores on the percent total SFA scores indicate better performance. Several other papers (Choi et al., 2021; Leblebici et al., 2021; Povedano et al., 2021; Jose et al., 2019) have also used percent scores for reporting results of the SHUEE, please see citations below. We have added these citations in the revised manuscript.
- Choi, J. Y., Rha, D. W., Kim, S. A., & Park, E. S. (2020). The dynamic thumb-in-palm pattern in children with spastic cerebral palsy and its effects on upper limb function. Children, 8(1), 17.
- Leblebici, G., Ovacik, U., Gungor, F., Davids, J. R., Tarakci, E., & Kasapcopur, O. (2021). Validity and reliability of “Shriners Hospital for Children Upper Extremity Evaluation” in children with rheumatic diseases. Clinical Rheumatology, 40(12), 5033-5040.
- Povedano, E., Gallardo-Calero, I., Navarrete, M., Adillon, C., Knorr, J., & Soldado, F. (2021). Analysis of dynamic elbow flexion deformity in children with hemiplegic cerebral palsy. Clinical Biomechanics, 81, 105245.
- Jose, P. S., Radhakrishna, V. N., Sahoo, B., & Madhuri, V. (2019). An assessment of the applicability of Shriners Hospital Upper Extremity Evaluation as a decision-making tool and outcome measure in upper limb cerebral palsy in Indian children. Indian Journal of Orthopaedics, 53, 15-19.
Comment 9: Note that the raw scores for the SHUEE are ordinal, and so the ANOVA and t-tests are not appropriate. What is the MCID for the SHUEE? A statistically-significant result is one thing, and although the ES is suggestive, these have been calculated as if the raw scores were interval-level data which they are not, and the MCID is important to determine if there was a difference that mattered to the children and their parents.
Response: We thank the reviewer for their thoughtful comment. Although SFA scores on individual items of the SHUEE are in the form of ordinal data, the total percent SFA and DPA scores calculated by summing SFA scores across all items of the SHUEE would be on the ratio scale. Our analyses are in lines with those conducted by other groups that also used percent scores on the SHUEE as outcome measures (please see citations above). However, we understand the reviewer’s point. Accordingly, we also conducted non-parametric tests (Wilcoxon signed rank tests) on outcome variables associated with the SHUEE and our results were similar (significant improvements in participants on the %SFA (z = -2.938; p = 0.003) and % unprompted affected UE use scores (z = -2.677; p = 0.007) and non-significant changes in the %DPA scores (z = -0.534; p = 0.594) and raw prompt scores (z = -1.841; p = 0.066)). Given the ease of interpretation of measures of central tendency and variability associated with parametric statistics, we report on data from parametric statistics. However, we confirm that all reported statistically significant findings were also significant with non-parametric tests. Moreover, all 11 children in the study demonstrated improvements in the % SFA scores and 9 out of 11 children improved on the %unprompted affected UE use scores from pretest to posttest. In the revised manuscript, we also provide individual data on the outcome measures associated with the SHUEE across all participants (please see Table 3). Although we agree with the reviewer that data on MCID values are invaluable to determine the clinical meaningfulness of the reported results, at present there are no data available in the literature for MCID of the SHUEE. However, in lines with our study, several other groups (please see citations below) have also used the SHUEE as an outcome measure for upper extremity interventions in CP. We have acknowledged the lack of availability of MCID values for the SHUEE in our revised manuscript (line 571-572, page 6.
- Lennon, N., Church, C., Shields, T., Kee, J., Henley, J. D., Salazar-Torres, J. J., ... & Ty, J. M. (2023). Can the Shriners Hospital Upper Extremity Evaluation (SHUEE) Detect Change in Dynamic Position and Spontaneous Function of the Upper Limb in People With Hemiplegic Cerebral Palsy? Journal of Pediatric Orthopaedics, 43(6), e471-e475.
- Van Heest, A. E., Bagley, A., Molitor, F., & James, M. A. (2015). Tendon transfer surgery in upper-extremity cerebral palsy is more effective than botulinum toxin injections or regular, ongoing therapy. JBJS, 97(7), 529-536.
Comment 10: I hope these comments will be taken as constructively as intended. I think there is mileage in intensive training camps such as the one for which you have demonstrated feasibility of incorporating Dual Joystick-operated ride-on-toy navigation Training. The benefits of such an inclusion remain to be demonstrated, and require an appropriate methodology that includes two or more randomly-allocated groups that are adequately-powered for the primary outcome measure.
Response: We thank the reviewer for their constructive comments. The edits made to the manuscript based on the reviewer’s suggestions have significantly strengthened the paper. Our ultimate aim with this line of work is to assess the utility of ride-on-toys as therapy adjuncts for children with CP across a variety of different settings (including camp, home, and school) and when delivered by a variety of stakeholders including clinicians, families, and teachers. As mentioned above, this was an unfunded pilot study to assess the feasibility of implementation and preliminary adjunctive effects of a training program using ride-on-toys. Our future study is a clinical trial where we will use a 2-group randomized controlled trial design to systematically compare ride-on-toy navigation training to a dose-matched program based on conventional therapy. We have ensured that we have highlighted the study design as a limitation for the present study and have also briefly discussed our future directions with this work in the limitations section of the revised manuscript (lines 1191-1198, page 13).
Reviewer 2 Report
Comments and Suggestions for Authors
Dear Author;
Your study presenting the importance of bilateral activity in children with UCP and a new rehabilitation approach is very interesting. However, there are some parts that I have difficulty understanding and would like you to edit.
Abstract
The abstract should be a total of about 200 words maximum. (your abstract has 213 words)
"a standardized motor test" is a very broad term. It suggests many tests. But as far as I understand, you mean the SHUEE test. If you write the name of the test, it will be more understandable.
Should a p-value be given for before and after evaluations?
Introduction
It is written in detail supported by the literature, but it is quite long.
The introduction should briefly place the study in a broad context and highlight why it is important.
Method
Abbreviations should be defined the first time. The abbreviation DJT was used twice.
MACS level can be added to the inclusion criteria.
“A total of 11 children participated in the study (4 males, 7 females; 6 children with 135 right hemiparesis and 5 children with left hemiparesis; Age in years – Mean (SE): 136 6.46(0.64)). Among the participating families, 8 families were Caucasian Non-Hispanic, 1 137 family was Caucasian Hispanic, and 2 families were multi-racial (one family mixed White 138 and Asian and one family mixed Korean, Puerto-Rican, Irish, and Polish). Children 139 demonstrated mild to moderate degree of impairment in their ability to use their hands 140 for daily activities (Manual Ability Classification System (MACS) scores: Mean (SE) – 141 2(0.19)).” Since this section expresses the results, it is appropriate to include it in the conclusion section.
In Figure 1, it is seen that the child's weight transfer and alignment are asymmetrical. It suggests that the quality of movement was not paid attention during the exercise.
Results
If demographic information and before-after comparisons are given in a table, it will be easier to understand.
What was the distribution of children according to MACS?
Discussion
There is a detailed literature review and commentary, but it is quite long. I suggest that some references older than 10 years should be removed and simplified.
As you mentioned, the absence of a control group makes it difficult to interpret the results. I wish you had a control group.
References
Please check the references.
References should be described as follows,
“1. Author 1, A.B.; Author 2, C.D. Title of the article. Abbreviated Journal Name Year, Volume, page range.”
There are 78 references in the text. But the 78th reference is not numbered in the references section.
49 of the 78 references are older than the last five years. It is preferable to use current references.
Tables and Figures
It will be more understandable if the results given in writing are tabulated.
The * in figures should be explained.
Author Response
We thank the reviewer for their thoughtful comments. Please see follow for our responses.
Comment 1: Abstract. The abstract should be a total of about 200 words maximum. (your abstract has 213 words) "a standardized motor test" is a very broad term. It suggests many tests. But as far as I understand, you mean the SHUEE test. If you write the name of the test, it will be more understandable. Should a p-value be given for before and after evaluations?
Response: We thank the reviewer for their comment. In the revised manuscript, we have edited the abstract to state the test being used in our study and have also provided p values for the reported significant results. In the revised manuscript, the abstract is exactly 200 words.
Comment 2: Introduction It is written in detail supported by the literature, but it is quite long.
The introduction should briefly place the study in a broad context and highlight why it is important.
Response: We appreciate the reviewer’s point. In the revised manuscript, we have significantly shortened the introduction (to around 1.25 pages instead of the previous and instead of the previous 1.75 pages) and instead directed the reader to relevant references in the area.
Comment 3: Method Abbreviations should be defined the first time. The abbreviation DJT was used twice.
Response: The acronym DJT has been first defined in the last paragraph of the introduction section (line 105) in the revised manuscript prior to the Methods section. We have checked the revised manuscript to ensure that all terms have been defined prior to use of their acronyms.
Comment 4: MACS level can be added to the inclusion criteria.
Response: For our study, we were open to working with children with varying MACS levels. We have added this information within the “Participants and Setting” section of the revised manuscript (lines 346-348, page 3).
Comment 5: “A total of 11 children participated in the study (4 males, 7 females; 6 children with 135 right hemiparesis and 5 children with left hemiparesis; Age in years – Mean (SE): 136 6.46(0.64)). Among the participating families, 8 families were Caucasian Non-Hispanic, 1 137 family was Caucasian Hispanic, and 2 families were multi-racial (one family mixed White 138 and Asian and one family mixed Korean, Puerto-Rican, Irish, and Polish). Children 139 demonstrated mild to moderate degree of impairment in their ability to use their hands 140 for daily activities (Manual Ability Classification System (MACS) scores: Mean (SE) – 141 2(0.19)).” Since this section expresses the results, it is appropriate to include it in the conclusion section.
Response: We are unsure of what the reviewer’s advice is in this comment. In the revised manuscript, per Reviewer 1’s suggestion, we have presented participant information in table 1 and have removed the detailed description of participants from the narrative text. We have also ensured that the conclusion section summarizes the demographics of the participants in this study.
Comment 6: In Figure 1, it is seen that the child's weight transfer and alignment are asymmetrical. It suggests that the quality of movement was not paid attention during the exercise.
Response: We understand the reviewer’s point. All researchers delivering the intervention sessions were physical therapists with experience working with children with cerebral palsy. When appropriate, we facilitated optimal alignment of the child’s trunk through verbal cues to encourage children to actively correct their position, tactile cues on the child’s trunk to facilitate weight shifts and more symmetrical weight bearing, and if needed manual therapist-initiated repositioning of the child’s body to facilitate symmetry, optimal alignment, and postural control. If necessary, we also used pool noodles to provide additional trunk support as required. We have edited Figure 1 and added this explanation in the revised manuscript to reflect our efforts of encouraging optimal positioning and symmetry in children while driving the toy (lines 478-484, page 4).
Comment 7: Results If demographic information and before-after comparisons are given in a table, it will be easier to understand.
Response: Per the suggestions of both reviewers, we have added Table 1 to list demographic characteristics of children. We have also added multiple tables (Tables 2-4) to provide individual data from all 11 children in the study to supplement the summary statistics and group data graphs presented in the paper.
Comment 8: What was the distribution of children according to MACS?
Response: Table 1 lists the MACS levels of all participants in the study.
Comment 9: Discussion: There is a detailed literature review and commentary, but it is quite long.
Response: Per the reviewer’s suggestions, we have significantly shortened the discussion section in the revised manuscript (from 2.5 pages in the previous version to 1.75 pages in the current version).
Comment 10: I suggest that some references older than 10 years should be removed and simplified.
Response: We understand the reviewer’s point. In the revised manuscript, as far as possible, we have removed references that are older than 10 years and have replaced some of these classical references with more recent evidence. Our edits have also reduced the number of references to 64 from the previous 78 references and simplified the paper per the suggestions of the reviewer.
Comment 11: As you mentioned, the absence of a control group makes it difficult to interpret the results. I wish you had a control group.
Response: We understand the reviewer’s point. This was a pilot unfunded study that limited our ability to conduct a 2-group study design. We have been careful throughout the manuscript to mention that the reported results may be attributed to the camp programming inclusive of the dual joystick ride-on-toy navigation training (DJT) program. Our future studies will involve a clinical trial with a two-group design that will compare the effects of DJT to a dose-matched program based on conventional therapy. We have added information of our future trial within the limitations section of the manuscript (lines 1191-1198, page 13).
Comment 12: References: Please check the references.
References should be described as follows,
“1. Author 1, A.B.; Author 2, C.D. Title of the article. Abbreviated Journal Name Year, Volume, page range.”
Response: In the revised manuscript, we have edited the references to ensure that they are in the designated format.
Comment 13: There are 78 references in the text. But the 78th reference is not numbered in the references section.
Response: We have corrected our typo in the revised manuscript. The revised paper has a total of 64 references.
Comment 14: 49 of the 78 references are older than the last five years. It is preferable to use current references.
Response: We understand the reviewer’s point. In the revised manuscript, as far as possible, we have used current references and removed references that are over 10 years old. We hope the reviewer will find our edits satisfactory.
Comment 15: Tables and Figures: It will be more understandable if the results given in writing are tabulated. The * in figures should be explained.
Response: Per both reviewers’ suggestions, we have added tables 2-4 that provide individual data from all 11 children seen in the study. We hope that these tables help clarify our results and aid interpretation. We have also added footnotes to explain the Asterix (*) for all the figures in the revised manuscript.
Round 2
Reviewer 1 Report
Comments and Suggestions for Authors
Dear authors
Thanks for making those changes, good luck with your randomised controlled trial.